# OpenReview forum: "Towards the Scalable Evaluation of Cooperativeness in Language Models"
_TMLR — Rejected by TMLR_

### Review · Reviewer_swc5 · 2023-04-11

**Summary Of Contributions:**

If my understanding is correct, this work is about evaluating the cooperativeness of language models, by using some pre-defined games, such as the dictator game (DG) and the ultimatum game (UG). The main contribution of this work consists of (1) formulating the evaluation with several scenarios in game theory; (2) creating text examples of these scenarios for evaluation.

**Audience:**

Yes

**Broader Impact Concerns:**

The broader impact statement is sufficient.

**Claims And Evidence:**

No

**Requested Changes:**

1. Please address the inconsistency of the storyline, and make sure this work is only about creating a dataset to evaluate the cooperativeness of language models, as indicted in the paper title
2. Please add necessary and also critical details to the paper, make sure the proposed method can be reproduced

**Strengths And Weaknesses:**

First, I have to say that I could not fully understand this work. It is not because of the lack of necessary background. On the contrary, I think this paper did a good job of explaining the concepts related to game theory. However, throughout the whole paper, I cannot find a coherent story. In other words, I think I neither get the research questions nor understand how these questions were answered in this work.

While reading the paper, sometimes I felt that this work is about using language models to help create some examples in game theory, and sometimes I thought this work is about using game theory to help understand language models.

For example

- In section 1, contribution 2 says "... and a language model have serious difficulty in both generating and judging the quality of evaluations that fit particular game-theoretic structures", which sounds like the research question is about "the quality of evaluation" instead of "evaluating language models"
- If the goal of this work is to create a dataset that can evaluate language models, then section 3.4 will create a circular issue of this work: a dataset was evaluated by language models to prove it is qualified to evaluate language models.


In addition, some important technical details are missing in the current draft, for example

- In section 2.2, "We developed the crowdworker questions after several cycles of iteration", which main issues were addressed by these "several cycles of iteration"?
- In section 3.4, "We finetuned GPT-3 ...", it's critical to explain how this was done
- In section 3.4, "We next tried few-shot chain-of-thought ...", again, it would be helpful to provide more description about how this was done.

---

### Review · Reviewer_xUu3 · 2023-04-24

**Summary Of Contributions:**

This paper focuses on an interesting and important domain of the cooperativeness of artificial intelligence agents, particularly pre-trained language models (PLMs). The authors present several key contributions to the field:

- They develop a methodology for generating evaluation scenarios that align with specific game-theoretic structures, applicable to both crowdworkers and language models.
- They discover that both human crowdworkers and language models face significant challenges in generating and judging the quality of scenarios adhering to the intended game-theoretic structures.
- They provide a dataset based on the generated data for further analysis and evaluation.
- They conduct quantitative and qualitative evaluations of UnifiedQA and GPT-3 models using the dataset, finding that larger instruct-tuned GPT-3 models tend to choose actions that could be viewed as cooperative, while other models exhibit flat scaling trends.

The paper demonstrates the difficulty and importance of generating diverse evaluation scenarios with clear game-theoretic structures to better understand and shape the multi-agent behaviors of PLMs in a pro-social manner.

**Audience:**

Yes

**Broader Impact Concerns:**

The following concerns could be further addressed or elaborated upon:

- The potential for biases in the generated dataset, as well as the implications of these biases on the evaluation and subsequent behavior of the AI systems.

- The potential consequences of relying heavily on game-theoretic structures in defining the evaluation scenarios, which may not always capture the complexity and nuance of real-world interactions and conflicts.

**Claims And Evidence:**

Yes

**Requested Changes:**

- Experiments with open-sourced models (critical): To increase the generalizability and accessibility of the findings, the authors should consider conducting experiments with open-sourced language models in addition to the proprietary models used in the paper.

- Reframe the motivation (critical): The motivation of the paper should be reframed to focus more on the practical and immediate challenges of fostering cooperative AI systems, reducing the emphasis on speculative scenarios.

- Illustrate more on how the dataset is filtered and report on data quality (critical): The authors should provide a more detailed explanation of the dataset filtering process and offer a clearer assessment of the data quality, including any issues encountered and potential biases.

-  Cite more relevant literature in NLP (e.g., Delphi: Towards Machine Ethics and Norms): To strengthen the work's position within the existing literature, the authors should cite and discuss more relevant papers in the field of NLP, such as the Delphi paper on machine ethics and norms, to contextualize their research and highlight its novelty.

**Strengths And Weaknesses:**

Strengths:
- The paper is innovative and forward-thinking, as it addresses the important issue of understanding and shaping the multi-agent behaviors of PLMs in a pro-social manner.
- The proposed methodology, if successful, could have a positive social impact by ensuring that AI systems act more cooperatively in high-stakes interactions.
- Introducing game theory as a framework for understanding the cooperativeness of AI agents is an interesting and novel approach.

Weaknesses:
- The motivation of the paper is largely speculative, relying on "maybe" scenarios about AI systems in high-stakes interactions.
- The quality of the generated dataset is hard to understand, and the size of the dataset is relatively small, which may limit the generalizability of the findings.
- As the capabilities of language models continue to improve, some of the findings in this paper may not hold, making the analysis somewhat time-sensitive.
- The paper does not provide much insight into the inner workings of the models being evaluated, focusing mainly on black-box models, which may limit the understanding of why certain behaviors arise.

---

### Review · Reviewer_ivur · 2023-04-27

**Summary Of Contributions:**

The paper focuses on understanding pre-trained LM behavior on cooperative scenarios. To investigate the problem, the author propose methodology for generating scenarios, by humans or pre-trained LMs, that correspond to thought experiments from game theory such as ultimatum, punishment, and prisoner's dilemma. The data collection shows that humans and pre-trained LMs have difficulty in generating accurate scenarios especially for prisoner's dilemma but humans are more accurate on ultimatum and punishment scenarios.

Using the curated data from the annotation, the authors then evaluate GPT models automatically on answering multiple-choice questions about the different scenarios and manually in an interactive setting. The results that instruct GPT models tend to perform better than non-instruct ones that were evaluated and their responses are more cooperative.

**Audience:**

Yes

**Broader Impact Concerns:**

The authors provide data & methods for evaluating cooperative capabilities of the models but there is a lack of discussion about if/how they can be misused to promote antagonistic/malicious behaviors and how they could be potentially mitigated.

Another aspect that is missing is discussing about the limitations of this work e.g. with regards to models being evaluated, settings, strength of claims, and scope of findings.

**Claims And Evidence:**

No

**Requested Changes:**

C1. Provide more information about what are the differences between different models that are being evaluated; not all the readers are familiar with different model codes and their differences. Some of the claims are hard to verify without this information.

C2. I suggest that the authors evaluate publicly available pre-trained **generative** models such as OPT, GPT-J, Llama, Alpaca. How do we know that the results are generalizable to other models and that there is no other confounding factor?

C3. Regarding the comparison of non-instruct models, what are the specific checkpoints that are being evaluated and why there appear to be differences between the scores obtained in Figure 8 and Figure 4? davinci models appear to be better than curie and other families in Figure 4 but it's not the case for Figure 8 which is very confusing. Also, it's unclear if the scores in these two are from the same experiment.

C4. It would be useful to include a persona that is intentionally antagonistic or unfair in a way that humans can be in certain scenarios. Can instruct GPT models be successfully prompted for such behavior?

C5. Claims in the introductions and abstract appear to be broad & bold which are not backed by rigorous evidence/experiments and I'd suggest the authors to better define the scope of their findings & make their claims specific.

**Strengths And Weaknesses:**

**Strengths**

Game-theoretic structures can manifest in end-user scenarios and, hence, evaluating pre-trained LMs in solving cooperation problems is an important direction as their availability and wide-spread adoption increases.

The idea of evaluating cooperative capabilities of LMs in games such as Ultimatum and Diplomacy has been explored in the past and this paper contributes data with higher diversity for different games and methods for automatically generating them.

Contributes a curated dataset with annotated scenarios for different games that can be used by the community for evaluation before deployment.

Finds that instruct GPT models tend to be more cooperative than non-instruct models which is a positive finding from an AI safety perspective.

**Weaknesses**

W1. The results are obtain with a few experimental settings and I am not convinced that we can rely on the conclusions drawn from them. Pre-trained LMs tend to be sensitive to different prompts and decoding strategies.

W2. Experiments are based on closed pre-trained generative LM models for which we know little about and lack verification of the results on publicly available generative models with specific architectures, training procedures and training data.

W3. The claim regarding scaling is confusing due to lack of details about which specific model checkpoints were used and is highly speculative since there is no information provided about the sizes of the models.

W4. Malicious use of pre-trained LMs to be intentionally non-cooperative or antagonistic has not been investigated while it seems important when talking about cooperation.

---

### Decision · Action_Editors · 2023-06-04

**Recommendation:** Reject

**Comment:**

Lack of evidence to support the claims

**Audience:**

The larger LM literature as models are used to respond to situations, but specifically those exploring the social role of use of LMs.

**Claims And Evidence:**

This work focuses on evaluating language models as agents/players in classic game-theory settings like Dictator and the Prisoner's dilemma.  The setup is paralleled for human players as well.

The authors claim: That both language models and humans have difficulty in generating/judging behaviors consistent with the behaviors detailed in the literature for these games.

They contribute:
1. Generation of new data that follows the classic relationships but in novel settings.  Synthetic settings are also included.
2. A qualitative evaluation of how models and humans differ in this setting.

There are several potentially interesting results (e.g. larger models' behavior changes when provided roleplaying prompts), however
while there is a substantial audience interested in the role of LLMs as social agents, the reviewers did not feel that the claims were supported. Specifically,

- Limited experimental settings with insufficient details, on the data creation and filtering, and the effect of scale
- The use of a closed LM makes it difficult to evaluate claims because there's a lack of comparisons to existing literature and the details of the model itself are not provided

Both of these prevent them being able to evaluate if there is sufficient evidence to support the claims. Note, that while there is a lot of useful content in the appendix it is not distilled clearly or explained in the main text.

There are also useful recommendations in the reviews about experiments to include (like non-cooperative agents) and the larger literature in which to reframe an updated story.

**Resubmission Of Major Revision:**

The authors may consider submitting a major revision at a later time.